# Difference between Female and Male Patients with Oral Squamous Cell Carcinoma: A Single-Center Retrospective Study in Taiwan

**DOI:** 10.3390/ijerph17113978

**Published:** 2020-06-04

**Authors:** Nan-Chin Lin, Jui-Ting Hsu, Kuo-Yang Tsai

**Affiliations:** 1School of Dentistry, China Medical University, Taichung 404, Taiwan; tifa92180@gmail.com (N.-C.L.); jthsu@mail.cmu.edu.tw (J.-T.H.); 2Department of Oral and Maxillofacial Surgery, Changhua Christian Hospital, Changhua 500, Taiwan; 3College of Nursing and Health Science, Da-Yeh University, Changhua 515, Taiwan

**Keywords:** head and neck cancer, lifestyle, squamous cell carcinoma of head and neck, survival analysis

## Abstract

There is a distinct male predominance in head and neck cancers. The present study aimed to investigate the clinical and pathological features of male and female patients with oral squamous cell carcinoma (OSCC), and to simultaneously conduct a survival analysis. Patients (*n* = 2573) were identified between January 2008 and December 2018, and subsequently analyzed for characteristics such as age at squamous cell carcinoma diagnosis, lifestyle factors (smoking habit, betel nut chewing and alcohol consumption), pathological American Joint Committee on Cancer (AJCC) anatomic site, AJCC TNM stage, pathological recurrence factor and interval from first diagnosis to recurrence. A case-matched comparison between female (*n* = 122) and male (*n* = 2451) patients was conducted. Significant gender differences were noted in age at diagnosis, anatomic site of the tumor, smoking habit, betel nut chewing and alcohol consumption (*p* < 0.001). There were no significant gender differences in the other clinical and pathological characteristics and survival conditions. In conclusion, female patients with OSCC were older than male patients with OSCC, and mostly had tumors of the oral tongue. Once patients develop OSCC, there was no difference in survival between men and women in a Taiwanese population.

## 1. Introduction

Gender-based differences in head and neck cancers (HNC) have been described in the literature, with a distinct male predominance. A study involving 703 patients with oral squamous cell carcinoma (OSCC) in southern Taiwan from 1985 and 1996 found a 51:1 male-to-female ratio [1], whereas other studies have reported OSCC in 90–93% male and 7–10% female patients [2,3,4]. This disparity in genders may be explained by the lower prevalence of some carcinogenic lifestyle factors, including smoking habits, betel nut chewing and alcohol consumption in women [5]. Concern about the disfiguring effects of these lifestyles factors (including red or brown staining of lips and teeth and foul-smelling breath) has been frequently reported by women, which may account for the sex-related differences in the prevalence of HNC [5].

OSCC accounts for 95% of all oral cancers and is associated with smoking habits and alcohol consumption in Western countries. Betel nut chewing and smoking are predominant risk factors causing OSCC in Taiwan and South Asia [1]. In Taiwan, approximately 2.5 million people are betel nut chewers. The incidence of OSCC was reportedly higher in betel nut chewers than in non-betel nut chewers [6]. A higher mortality rate among patients with OSCC was found to be associated with the increasing prevalence of betel nut chewing [6].

Feghali et al. reported that among patients with OSCC, heavy smokers have more aggressive pathological features and diseases than non-smokers [7]. It would be of interest to investigate whether there are differences in the clinical and pathological characteristics and survival conditions between male and female patients with OSCC, based on differences in carcinogenic lifestyle factors. The present study aimed to investigate the clinical and pathological characteristics of male and female patients, and to conduct survival analysis simultaneously.

## 2. Materials and Methods

### 2.1. Patients

This retrospective cohort study was approved by the Institutional Review Board (IRB) and Ethical Committee of Changhua Christian Hospital (IRB number 190408). All clinical data were obtained through review of medical charts and through the cancer registry center. A total of 3248 patients diagnosed with oral cavity and lip squamous cell carcinoma (SCC), who received treatment and follow-up at the Changhua Christian Hospital between 1 January 2008 and 31 December 2018, were identified. Patients were followed up until 30 June 2019. Patients were excluded for the following reasons: not receiving treatment per the American Joint Committee on Cancer (AJCC) cancer treatment guidelines (*n* = 41), lip cancer not involving mucosa (*n* = 121), lost to follow-up or incomplete data (*n* = 102), initially diagnosed with recurrent or distant metastasis (*n* = 70), or not undergoing surgery at Changhua Christian Hospital (*n* = 341). Finally, 2571 patients were identified and subsequently analyzed. A case-matched comparison of clinical and pathological characteristics between female (*n* = 122) and male (*n* = 2451) patients was conducted.

### 2.2. Clinical Parameters

The analyzed characteristics of both the groups included age at SCC diagnosis, lifestyle factors (smoking habits, betel nut chewing and alcohol consumption), pathological AJCC anatomic site, AJCC TNM stage, pathological recurrence factor and interval from first SCC diagnosis to recurrence. The anatomic sites were subclassified into alveolar ridge, anterior two-third tongue, buccal mucosa, hard palate, floor of the mouth, retromolar trigone and lip mainly involving the mucosa. Pathological recurrence factors included extranodal spread, close margin (<1 mm or involved margin) and type of tumor cell differentiation. Death information was retrieved from the cancer registry center of Changhua Christian Hospital, and the data were renewed annually by the Health Bureau of Changhua city.

We designed an observational, retrospective study based on gender, to compare the clinical and pathological characteristics and conducted a survival analysis between the groups.

### 2.3. Statistical Analysis

Continuous variables are presented as mean ± standard deviation (SD), and categorical variables are presented as percentages. Student’s t-test was used to compare continuous variables between cases and controls and the chi-squared test was used to compare differences in categorical variables between the two groups. Estimates of the overall survival (OS) rates were calculated using Kaplan–Meier analyses. Comparisons of group survival functions were conducted using log-rank tests based on OS rates. A p-value of <0.05 was considered statistically significant. All statistical analyses were performed with the statistical package SPSS for Windows (Version 16, IBM, Chicago, IL, USA).

## 3. Results

Our retrospective study enrolled 2573 patients; 122 women as cases and 2451 men as controls. The clinical and pathological characteristics of the patients are summarized in Table 1 and Table 2.

Table 1 summarizes the age at diagnosis, survival time (with a follow-up duration from index data to 30 June 2019), and interval from diagnosis to recurrence in all patients. There was a significant difference between men and women in the mean age at diagnosis of cases (*p* < 0.001). Among cases, the mean age at diagnosis was 61.7 years, whereas it was 56.9 years in controls. There was no significant gender difference in survival time and interval from diagnosis to recurrence.

Table 2 summarizes the data stratified by age at diagnosis and the clinical and pathological characteristics of all patients. In terms of age at diagnosis, the majority of cases were diagnosed in Group 5 (>71 years), followed by Group 3 (51–60 years), whereas the majority of controls were diagnosed in Group 3 (51–60 years). There was a significant difference between cases and controls.

There was a significant difference between men and women (controls and cases) in terms of the anatomic site of the tumor. More than 50% of the cases had tumors of the anterior two-thirds of the tongue, whereas the majority of controls were most likely to have tumors of the buccal mucosa, followed by anterior two-thirds of the tongue.

Smoking habits, betel nut chewing and alcohol consumption were recorded in a significantly lower proportion in women (cases) than in men (controls). There was no significant gender difference in T status, N status or AJCC stage. With regard to recurrence status and recurrence factor, there was no significant gender difference in recurrence, extranodal spread, close margin or tumor differentiation grade.

The Kaplan–Meier curve indicated no significant gender difference in OS among all AJCC stages (*p* = 0.563), early stage (*p* = 0.689) or advanced stage (*p* = 0.166) subgroups between cases and controls (Figure 1, Figure 2 and Figure 3).

## 4. Discussion

In the present study, the most notable differences between females and males patients with OSCC were found in terms of age at diagnosis and anatomic site of tumors. Older women (>70 years) were more affected than younger women, suggesting an effect of cumulative opportunity to develop OSCC. Male patients with OSCC were mainly in the age group of 51–60 years. Most female patients had tumors of the oral tongue, whereas most male patients had tumors of the buccal mucosa. In addition, male patients had distinct carcinogenic habits, including smoking, betel nut chewing and alcohol consumption. There were no differences in OS between male and female patients, suggesting that once patients have OSCC, they have similar survival conditions, regardless of gender.

Several studies have established that alcohol consumption, cigarette smoking and betel nut chewing contribute to an increased incidence of HNC [8,9,10,11]. These lifestyle factors were prominently found in the male group, which could explain the notable earlier age at diagnosis of OSCC in male patients, compared with that observed in female patients. Kruse et al. also reported that the median age at OSCC diagnosis was higher in female patients than in male patients (65.36 vs. 61.04 years), with a predominance of female patients older than 70 years, similar to that observed in the present study [12].

In contrast, the most common anatomic site of tumors among female patients in Kruse’s study was the gum, while, in our study, it was the tongue. This result of the present study is similar to those of other studies reported around the world [13,14]. Ng et al. demonstrated a general trend of increasing incidence of oral tongue cancer which appears to hold true across most registries [14]. These findings suggest the possible emergence of new etiological or genetic factors driving carcinogenesis in the tongue, which features more strongly in certain geographic areas and in different sexes, accounting for the heterogeneous results observed in the present study [15]. Annertz et al. indicated a continued increasing trend in the incidence of OSCC of the tongue in the 21st century for both the sexes and all age groups, except for young male patients; however, the cause of this trend is controversial [13]. In the present study, the risk factors contributing to tongue OSCC in female patients have not been addressed. The non-smoking, non-drinking and non-betel nuts chewing patients with OSCC may be associated with other etiologies and risk factors. Poor oral hygiene, inadequate dental status and chronic irritation are independent risk factors for OSCC, irrespective of tobacco and alcohol consumption [16]. Human papillomavirus (HPV) has been implicated as a major risk factor in oropharyngeal cancer; however, the prevalence of high-risk HPV in cancer of the oral tongue appears to be low [17,18,19]. None of the known risk factors appear to explain the increasing incidence of OSCC of the oral tongue. Further research is warranted to investigate this issue in the future.

Betel nut chewing is very common lifestyle factor in Taiwan, in particular among blue-collar workers, men, low socioeconomic status people or indigenous people [6]. Betel nut chewers have been shown to have a lower median age at onset (approximately 6–12 years earlier) than non-chewers [20], and this finding could explain why more men were diagnosed with OSCC at a younger age than women in the present study. Furthermore, Su et al. reported that the high incidence of the buccal mucosa, as the site of OSCC may have a strong association with betel nut chewing among male patients [20]. In Japan, betel nut chewing is not as common as in Taiwan. Koyama et al. reported that from 2000 to 2014, a total of 6086 cases of oral cavity cancer were identified. Among all the patients, approximately 40% were women, and tongue cancer accounted for 40% of cases, whereas gum cancer accounted for 32%; buccal mucosa cancer accounted for a small proportion of all cases [21]. Thus, it can be concluded that in OSCC cases, the buccal mucosa is the predominant lesion sites among betel nut chewers. There was no difference in OS between female and male patients with OSCC in our study, although male patients had a distinct lifestyle leading to OSCC. Giraldi et al. reported that the status of pre-diagnosis cigarette smoking and alcohol consumption were not prognostic factors for head and neck specific survival in OSCC, but smoking more than 20 cigarettes per day affected OS among patients with OSCC [22]. On the other hand, Kawakita et al. demonstrated that pre-treatment smoking status did have a prognostic value in patients with OSCC, but this effect was evident only among radiotherapy (RT) or concurrent chemoradiotherapy (CCRT) groups; they concluded that smoking may affect RT or CCRT treatment [23]. In the present day, surgery is still the mainstay treatment in patients with OSCC.

There are several limitations to our study. First, the data of our study were collected from a single medical center in Taiwan, and the people in a single district may have some cultural and geographical features. Second, we did not quantify lifestyle factors—smoking, betel nut chewing and alcohol consumption—and collected the data on these lifestyle factors after the patients had received cancer treatment, which may affect the survival condition. Moreover, a considerable amount of data on lifestyle factors were not available. Although the results were not affected due to distinct gender differences in lifestyle factors, further data analysis could be impeded. Finally, the relatively small sample size of women limited the capacity for further stratified analysis.

## 5. Conclusions

In conclusion, female patients with OSCC were older than male patients and mostly had tumors on the oral tongue. Once the patients had OSCC, there was no difference in OS between male and female patients in this Taiwanese population.

## Figures and Tables

**Figure 1 ijerph-17-03978-f001:**
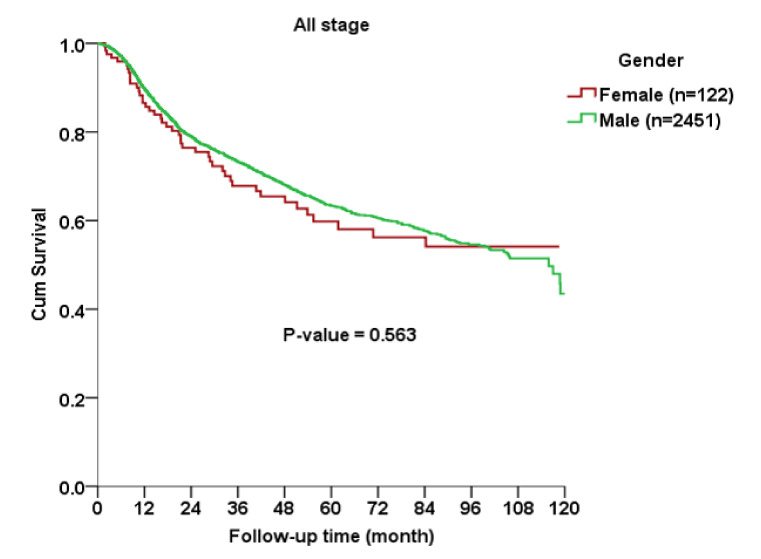
Kaplan–Meier curve for overall survival rates between female and male patients with oral squamous cell carcinoma.

**Figure 2 ijerph-17-03978-f002:**
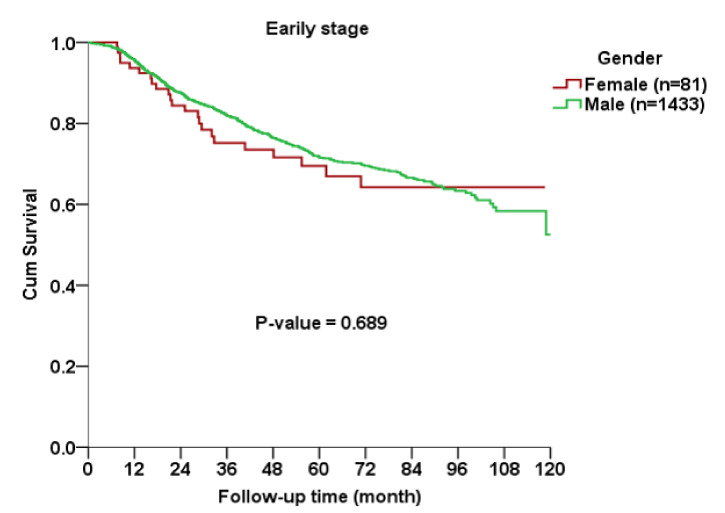
Kaplan–Meier curve for overall survival rates between female and male patients with early-stage oral squamous cell carcinoma.

**Figure 3 ijerph-17-03978-f003:**
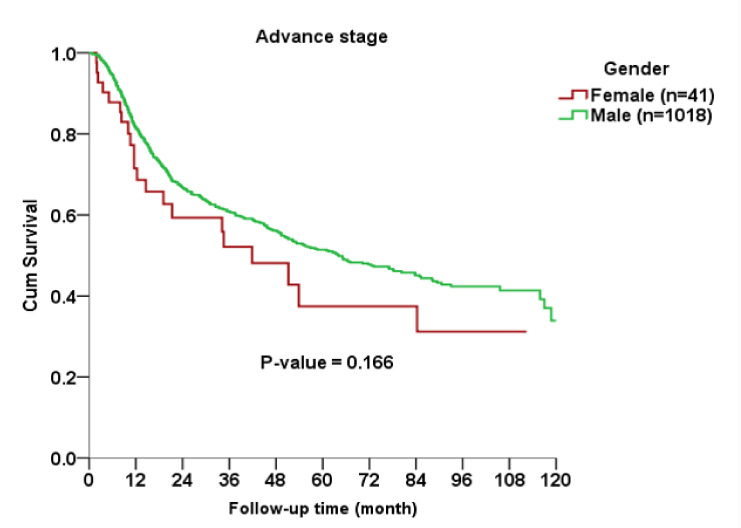
Kaplan–Meier curve for overall survival rates between female and male patients with advanced-stage oral squamous cell carcinoma.

**Table 1 ijerph-17-03978-t001:** Age at diagnosis, survival time (follow-up duration was from index data to 30 June 2019), and interval from diagnosis to recurrence in all patients with oral squamous cell carcinoma (OSCC); *p-*value was tested by Student’s *t*-test.

Clinical Characteristics	Gender						Total			*p*-Value
	Female			Male						
	n	Mean	SD	*n*	Mean	SD	*n*	Mean	SD	
Age at diagnosis (years)	122	61.7	15.1	2451	56.9	10.9	2573	57.2	11.1	0.001
Survival time * (months)	122	45.4	33.8	2451	44.7	31.0	2573	44.7	31.1	0.824
Time from diagnosis to recurrence (months)	30	11.6	10.9	484	13.2	12.6	514	13.1	12.5	0.496

* Survival time was calculated from OSCC diagnosis to death or June 30, 2019.

**Table 2 ijerph-17-03978-t002:** Stratified data including age at diagnosis and clinical and pathological characteristics of all patients; AJCC: American Joint Committee on Cancer; *p*-value was tested by chi-square test.

Clinical and Pathological Characteristics		Gender	Total(*n* = 2573)	*p*-Value
Female(*n* = 122)	Male(*n* = 2451)
*n*	%	*n*	%	n	%	
Age (years)	≤40	12	9.8	153	6.2	165	6.4	<0.001
	41–50	14	11.5	547	22.3	561	21.8	
	51–60	33	27.0	882	36.0	915	35.6	
	61–70	18	14.8	584	23.8	602	23.4	
	≥71	45	36.9	285	11.6	330	12.8	
AJCC anatomic site	Alveolar ridge	18	14.8	401	16.4	419	16.3	<0.001
	Anterior tongue	64	52.5	669	27.3	733	28.5	
	Buccal mucosa	23	18.9	881	35.9	904	35.1	
	Hard palate	1	0.8	73	3.0	74	2.9	
	Floor of the mouth	1	0.8	76	3.1	77	3.0	
	Retromolar trigone	4	3.3	133	5.4	137	5.3	
	Mucosa of the Lips	11	9.0	218	8.9	229	8.9	
Smoking	No	96	78.7	454	22.5	550	25.7	<0.001
	Yes	26	21.3	1562	77.5	1588	74.3	
	Unknown	0		435				
Betel nut chewing	No	111	91.0	808	40.1	919	43.0	<0.001
	Yes	11	9.0	1208	59.9	1219	57.0	
	Unknown	0		435				
Alcohol	No	116	95.1	844	47.5	960	50.6	<0.001
	Yes	6	4.9	931	52.5	937	49.4	
	Unknown	0		676				
AJCC Stage	I	62	50.8	942	39.3	1004	39.8	0.089
	II	19	15.6	444	18.5	463	18.4	
	III	9	7.4	207	8.6	216	8.6	
	IV	32	26.2	805	33.6	837	33.2	
AJCC Stage	Early (I and II)	81	66.4	1433	58.5	1514	58.8	0.082
	Advance (III and IV)	41	33.6	1018	41.5	1059	41.2	
T stage	1	66	54.1	1064	43.6	1130	43.9	0.129
	2	26	21.3	578	23.6	604	23.5	
	3	5	4.1	148	6.0	153	5.9	
	4	25	20.5	653	26.7	678	26.4	
N stage	0	62	70.5	1368	73.0	1430	72.9	0.247
	1	13	14.8	170	9.1	183	9.3	
	2	11	12.5	308	16.4	319	16.3	
	3	2	2.3	28	1.5	30	1.5	
Extranodal spread	No	76	84.4	1539	85.4	1615	85.4	0.801
	Yes	14	15.6	263	14.6	277	14.6	
Close margin	No	115	94.3	2368	96.6	2483	96.5	0.197
	Yes	7	5.7	83	3.4	90	3.5	
Recurrence	No	92	75.4	1967	80.3	2059	80.0	0.192
	Yes	30	24.6	484	19.7	514	20.0	
Grade	Well	12	10.1	423	17.5	435	17.2	0.076
	Moderately	98	82.4	1866	77.2	1964	77.5	
	Poor	9	7.6	127	5.3	136	5.4

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
