# Peer review of "Difference between Female and Male Patients with Oral Squamous Cell Carcinoma: A Single-Center Retrospective Study in Taiwan"

_ijerph, 2020, doi:10.3390/ijerph17113978_

Round 1

Reviewer 1 Report

This is a study mapping clinical differences between males and females with oral squamous carcinoma in Tawan.

The manuscript has an interesting subject, however, it is hastily written, and has many wrong facts. Already in the Abstract the number of patients included are wrong: "Patients (n=2571), females (n=122) and males (n=2573)". That makes a total of 2695 patients!!!

The same is wrong in Materials and Methods!!

As lip cancer has a completely different cause, it should not be mixed with oral tumours, and thus be taken out from the study.

In table 2 data on smoking, alcohol etc are nor available for some of the male patients, but a line with "unknown" must then be included. How was the significance calculated for these factors, when data for a considerable number of patients are missing??

The manuscript needs thorough and careful re-analysis and re-writing!

Author Response

Response letter

Tittle: The difference between female and male oral squamous cell carcinoma patients: a single-center retrospective case-control study in Taiwan

General response:

We sincerely thank the editor and all reviewers for their valuable feedback that we have used to improve the quality of our manuscript. The reviewer comments are laid out below in italicized font and specific concerns have been numbered. Our response is given in normal font and changes/additions to the manuscript are given in ”Track changes”.

Point-to-point response:

Reviewer #1:

  1. The manuscript has an interesting subject, however, it is hastily written, and has many wrong facts. Already in the Abstract the number of patients included are wrong: "Patients (n=2571), females (n=122) and males (n=2573)". That makes a total of 2695 patients!!!

Response: Thank you very much for your comments. We apologize for our negligence. We have made change and overall check for the manuscript. The change can be found on line 13, 18 and 65.

  1. The same is wrong in Materials and Methods!!

Response: Thank you very much for your comments. We apologize for our negligence. Our retrospective study enrolled 2573 patients, including 122 females as cases and 2451 males as controls. We have made change on line 65.

  1. As lip cancer has a completely different cause, it should not be mixed with oral tumours, and thus be taken out from the study.

Response: In this study, we only enrolled those patients with SCC on mucosa of lip. We have excluded those patients with cutaneous squamous cell carcinoma of the vermilion lip from the very first (line 61 to line 62, in the Material and Methods).  For better understanding, we have made change on Table 2 to prevent misunderstanding. Thank you for the comment.

  1. In table 2 data on smoking, alcohol etc are nor available for some of the male patients, but a line with "unknown" must then be included. How was the significance calculated for these factors, when data for a considerable number of patients are missing??

Response:

Thank you very much for your suggestion. I have added column “Unknown” in the Table 2, into lifestyle behavior section.

In the fact, the difference in lifestyle between females and males is very distinct. Even though pooling data of “Unknown” into “No” , the result has still not changed.

Gender

Female
(n=122)

Male
(n=2451)

Total
(n=2573)

P-value

N

%

N

%

N

%

Smoking

No + Unknow

96

78.7

889

36.3

550

38.3

<0.001

Yes

26

21.3

1562

63.7

1588

61.7

Betel nut

No + Unknow

111

91.0

1243

50.7

1354

52.6

<0.001

Yes

11

9.0

1208

49.3

1219

47.4

Alcohol

No + Unknow

116

95.1

1520

47.5

1636

63.6

<0.001

Yes

6

4.9

931

52.5

937

36.4

P-value by Chi-square test

Smoking

No

96

78.7

454

22.5

550

25.7

<0.001

Yes

26

21.3

1562

77.5

1588

74.3

Unknown

0

435

Betel nut

No

111

91.0

808

40.1

919

43.0

<0.001

Yes

11

9.0

1208

59.9

1219

57.0

Unknown

0

435

Alcohol

No

116

95.1

844

47.5

960

50.6

<0.001

Yes

6

4.9

931

52.5

937

49.4

Unknown

0

676

P-value by Chi-square test

However, missing data will not be discussed in the Table 2, and the results are not affected whether those missing data in or out.  We have added the drawback about considerable missing data in the Discussion, line 190 to 192. Thank you again for your valuable suggestions to improve the quality of our manuscript.

Reviewer 2 Report

Comments and Suggestions for Authors

Although this is an interesting article, there are some aspects that need revision:

Abstract: In line 8, I think that there is an error, as in results males are n=2451.

Introduction:

-second paragraph (lines 35-44) should be removed or discussed in “Discussion”, and also explain why women are more susceptible to lung cancer but prevalence is lower.

-in the third paragraph (lines 46-47) the last sentence should be removed as at the beginning of the introduction the authors have provide an article that shows that this sentence is not true.

Materials and Methods:

-in line 60 the same mistake as in the abstract should be corrected: male (n=2451) instead of (n=2573).

-Although men suffer more for oral cancer than women, case control study is not well used in this case. This is a prevalence study and authors are analysing clinicopathological features.  I strongly recommend to remove case-control references in methodology.

Results:

-In table 1 Legend (line 83), survival time unit (I guess that are months), should be added.

-in table 1, I think that also survival status has to be removed, as it is not interesting for the article.

-in table 1, (footnotes about p-value, AJCC should be moved to Legend in line 83.

-Line 95-96: “Age at diagnosis was divided into five…” this sentence is not interesting as reader can see it in the table, and it is a mere description.

Discussion:

-The Discussion should begin with a brief summary of highlighted results.

-First sentence in line 120 is not true (see article PMID 3227488).

-Line 121: “We found several differences...” several is not an accurate concept, be more precise.

-Lines 143-147 seem to be written without a logical order, these aspects must be contextualized and justified to be referred throughout the text

-Line 151, “Betel chewers have been shown to have lower median age of onset than non-chewers… and this funding coincided with our study.” I have not found the analysis of association between betel chewers and age of diagnosis, if this analysis has been already done a p-value should be attached.

Author Response

Response letter

Tittle: The difference between female and male oral squamous cell carcinoma patients: a single-center retrospective case-control study in Taiwan

General response:

We sincerely thank the editor and all reviewers for their valuable feedback that we have used to improve the quality of our manuscript. The reviewer comments are laid out below in italicized font and specific concerns have been numbered. Our response is given in normal font and changes/additions to the manuscript are given in ”Track changes”.

Point-to-point response:

Reviewer #2:

Abstract

  1. In line 8, I think that there is an error, as in results males are n=2451.

Response: Thank you very much for your comments. We apologize for our negligence. We have made change and overall check the manuscript. Our retrospective study enrolled 2573 patients, including 122 females as cases and 2451 males as controls. The change can be found on line 13 and 18.

Introduction

  1. Second paragraph (lines 35-44) should be removed or discussed in “Discussion”, and also explain why women are more susceptible to lung cancer but prevalence is lower.

Response: Thank you very much for your suggestion. It is indeed that second paragraph (lines 35 to 34) were not suitable discussed in the present study, and we did not addressed this part further in the Discussion. Based on this reason, we have revised second paragraph accordingly. The change can be found on line 45 to 48.

  1. In the third paragraph (lines 46-47) the last sentence should be removed as at the beginning of the introduction the authors have provide an article that shows that this sentence is not true.

Response: Thank you very much for your suggestion. We have removed the sentence accordingly. The change can be found on line 50 to 51.

Materials and Methods

  1. In line 60 the same mistake as in the abstract should be corrected: male (n=2451) instead of (n=2573).

Response: Thank you very much for your suggestion. We apologize for our negligence. Our retrospective study enrolled 2573 patients, including 122 females as cases and 2451 males as controls. We have made change on line 65.

  1. Although men suffer more for oral cancer than women, case control study is not well used in this case. This is a prevalence study and authors are analysing clinicopathological features. I strongly recommend to remove case-control references in methodology.

Response: Thank you very much for your comments. We have revised the tittle and manuscript accordingly. The change can be found on line 4 and line 77.

Results

  1. In table 1 Legend (line 83), survival time unit (I guess that are months), should be added.

Response: Thank you very much for your suggestion. I have revised the table 1 accordingly.

  1. In table 1, I think that also survival status has to be removed, as it is not interesting for the article.

Response: Thank you very much for your suggestion. I have removed survival status in the table 2 accordingly.

  1. In table 1, (footnotes about p-value, AJCC should be moved to Legend in line 83.

Response: I have moved footnotes about p-value and AJCC to table legend accordingly.

  1. Line 95-96: “Age at diagnosis was divided into five…” this sentence is not interesting as reader can see it in the table, and it is a mere description.

Response: Thank you very much for your suggestion. I have removed the sentence accordingly, and the change can be found on line 108 to 109.

Discussion

  1. The Discussion should begin with a brief summary of highlighted results.

Response: Thank you very much for your suggestion. I have revised the Discussion paragraph, and the change can be found on line 135 to 138.

  1. First sentence in line 120 is not true (see article PMID 3227488).

Response: Thank you very much for your suggestion. I have remove the first sentence and revised the Discussion. The change can be found on line 135 to line 138.

  1. Line 121: “We found several differences...” several is not an accurate concept, be more precise.

Response: Thank you very much for your suggestion. I have revised the first paragraph in Discussion, and directly pointed out the main objective finding in the present study. The change can be found on line 135 to 138.

  1. Lines 143-147 seem to be written without a logical order, these aspects must be contextualized and justified to be referred throughout the text

Response: Thank you very much for your suggestion. For better understanding, I have revised the manuscript. The change can be found on line 159 to 166.

  1. Line 151, “Betel chewers have been shown to have lower median age of onset than non-chewers… and this funding coincided with our study.” I have not found the analysis of association between betel chewers and age of diagnosis, if this analysis has been already done a p-value should be attached.

Response: Thank you very much for your suggestion. Actually, I would like to cite the finding from Su et. al (reference 19) to explain objective finding in our study. I have revised the manuscript for better understanding. The change can be found on line 168 to 171. Thanks again for your professional review work.

Reviewer 3 Report

This study descrived to investigate about the sex differences in oral cavity SCC of Taiwanese population. I think that the results were very interesting and the fact is quite important for particulaly ordinary people.

I think that this study would improve more interesting with the comparizon between other relatively near asian countries which does not have the life style of betel nut chewing, like Korea, Japan etc. These countries usually do not have the betel nut chewing life style. 

Maybe it is difficult to comparizen direct, could you please describe in  discussion section with certain data? 

Author Response

Response letter

Tittle: The difference between female and male oral squamous cell carcinoma patients: a single-center retrospective case-control study in Taiwan

General response:

We sincerely thank the editor and all reviewers for their valuable feedback that we have used to improve the quality of our manuscript. The reviewer comments are laid out below in italicized font and specific concerns have been numbered. Our response is given in normal font and changes/additions to the manuscript are given in ”Track changes”.

Point-to-point response:

Reviewer #3:

  1. I think that this study would improve more interesting with the comparizon between other relatively near asian countries which does not have the life style of betel nut chewing, like Korea, Japan etc. These countries usually do not have the betel nut chewing life style. Maybe it is difficult to comparizen direct, could you please describe in discussion section with certain data?

Response: Thank you very much for your positive comments. I have added the paragraph to describe oral cavity cancer data from Japan. The change can be found on line 173 to 177.

Round 2

Reviewer 1 Report

The authors have changed and corrected the ma usrciot.

However, the manuscript must go through grammatical editing including language corrections.

Author Response

Response letter

Tittle: The difference between female and male oral squamous cell carcinoma patients: a single-center retrospective study in Taiwan

General response:

We sincerely thank the editor and all reviewers for their valuable feedback that we have used to improve the quality of our manuscript. The reviewer comments are laid out below in italicized font and specific concerns have been numbered. Our response is given in normal font and changes/additions to the manuscript are given in ”Track changes”.

Point-to-point response:

Reviewer #1:

  1. The authors have changed and corrected the manuscript. However, the manuscript must go through grammatical editing including language corrections.

Response: Thank you very much for your comments and suggestion. We have re-send the manuscript to Enago for an English language review after your suggestion.

We have revised the manuscript extensively based on the reviewer’s comments. If there are any other modifications we could make, we would like very much to do so and we greatly appreciate your help. We hope that our manuscript will be considered for publication in your journal. Thank you very much.

Sincerely,

Kuo-Yang Tsai

Department of Oral and Maxillofacial Surgery, Changhua Christian Hospital

No. 235, Xuguang Rd, Changhua City, Changhua County 500, Taiwan

Phone: +886 933127916

Email: 72837@cch.org.tw

Reviewer 2 Report

Changes have improved significantly the text, although I still think introduction is too short. I recommend to write a paragraph to talk about oral squamous cell carcinoma features and the current situation of patients diagnosed with OSCC.

Author Response

Response letter

Tittle: The difference between female and male oral squamous cell carcinoma patients: a single-center retrospective study in Taiwan

General response:

We sincerely thank the editor and all reviewers for their valuable feedback that we have used to improve the quality of our manuscript. The reviewer comments are laid out below in italicized font and specific concerns have been numbered. Our response is given in normal font and changes/additions to the manuscript are given in ”Track changes”.

Point-to-point response:

Reviewer #2:

  1. Changes have improved significantly the text, although I still think introduction is too short. I recommend to write a paragraph to talk about oral squamous cell carcinoma features and the current situation of patients diagnosed with OSCC.

Response: Thank you very much for your comments. I have added a paragraph into Introduction accordingly, the change can be found in line 39 to 44 .

We have revised the manuscript extensively based on the reviewer’s comments. If there are any other modifications we could make, we would like very much to do so and we greatly appreciate your help. We hope that our manuscript will be considered for publication in your journal. Thank you very much.

Sincerely,

Kuo-Yang Tsai

Department of Oral and Maxillofacial Surgery, Changhua Christian Hospital

No. 235, Xuguang Rd, Changhua City, Changhua County 500, Taiwan

Phone: +886 933127916

Email: 72837@cch.org.tw